# Dual Targeting Ligands—Histamine H_3_ Receptor Ligands with Monoamine Oxidase B Inhibitory Activity—In Vitro and In Vivo Evaluation

**DOI:** 10.3390/pharmaceutics14102187

**Published:** 2022-10-13

**Authors:** Dorota Łażewska, Agata Siwek, Agnieszka Olejarz-Maciej, Agata Doroz-Płonka, Anna Wiktorowska-Owczarek, Marta Jóźwiak-Bębenista, David Reiner-Link, Annika Frank, Wioletta Sromek-Trzaskowska, Ewelina Honkisz-Orzechowska, Ewelina Królicka, Holger Stark, Marek Wieczorek, Waldemar Wagner, Katarzyna Kieć-Kononowicz, Anna Stasiak

**Affiliations:** 1Department of Technology and Biotechnology of Drugs, Faculty of Pharmacy, Jagiellonian University Medical College in Kraków, Medyczna 9 Str., 30-688 Kraków, Poland; 2Department of Pharmacobiology, Faculty of Pharmacy, Jagiellonian University Medical College in Kraków, Medyczna 9 Str, 30-688 Kraków, Poland; 3Department of Pharmacology and Toxicology, Medical University of Lodz, Żeligowskiego 7/9 Str., 90-752 Łódź, Poland; 4Institute of Pharmaceutical and Medicinal Chemistry, Heinrich Heine University Düsseldorf, Universitaetsstr. 1, 40225 Düsseldorf, Germany; 5Department of Neurobiology, Faculty of Biology and Environmental Protection, University of Lodz, Pomorska 141/143 Str., 90-236 Łódź, Poland; 6Department of Hormone Biochemistry, Medical University of Lodz, Żeligowskiego 7/9 Str., 90-752 Łódź, Poland; 7Laboratory of Cellular Immunology, Institute of Medical Biology of Polish Academy of Sciences, 106 Lodowa Str., 93-232 Łódź, Poland

**Keywords:** histamine H_3_ receptor, histamine H_3_ receptor ligand, monoamine oxidase B (MAO B), MAO B inhibitor, dual-target ligands, pitolisant, in vivo studies

## Abstract

The clinical symptoms of Parkinson’s disease (PD) appear when dopamine (DA) concentrations in the striatum drops to around 20%. Simultaneous inhibitory effects on histamine H_3_ receptor (H_3_R) and MAO B can increase DA levels in the brain. A series of compounds was designed and tested in vitro for human H_3_R (*h*H_3_R) affinity and inhibitory activity to human MAO B (hMAO B). Results showed different activity of the compounds towards the two biological targets. Most compounds had poor affinity for *h*H_3_R (*K_i_* > 500 nM), but very good inhibitory potency for hMAO B (IC_50_ < 50 nM). After further in vitro testing (modality of MAO B inhibition, permeability in PAMPA assay, cytotoxicity on human astrocyte cell lines), the most promising dual-acting ligand, 1-(3-(4-(tert-butyl)phenoxy)propyl)-2-methylpyrrolidine (**13**: *h*H_3_R: *K_i_* = 25 nM; hMAO B IC_50_ = 4 nM) was selected for in vivo evaluation. Studies in rats of compound **13**, in a dose of 3 mg/kg of body mass, confirmed its antagonistic effects for H_3_R (decline in food and a water consumption), decline in MAO B activity (>90%) in rat cerebral cortex (CTX), and an increase in DA content in CTX and striatum. Moreover, compound **13** caused a slight increase in noradrenaline, but a reduction in serotonin concentration in CTX. Thus, compound **13** is a promising dual-active ligand for the potential treatment of PD although further studies are needed to confirm this.

## 1. Introduction

Parkinson’s disease (PD) is characterized by a progressive loss of dopaminergic neurons in the *substantia nigra* and the accumulation of misfolded and aggregated α-synuclein named Lewy bodies. All of this leads to a decrease in the level of dopamine (DA) in the *striatum* causing memory deficits and also problems with moving. However, it should be remembered that a decline in DA levels is a normal process of ageing and it could be reduced by 40–50% of the beginning amount at the age of 60. However, the motor symptoms of PD (such as bradykinesia, tremor and stiffness) are observed when DA concentration is diminished by 80% of the initial volume. Compensation for DA shortage in the brain could be achieved among others by activation of DA receptors by the precursor of DA—*levodopa*, and DA agonists (e.g., *bromocriptine*, *rotigotine*), and by blocking of enzymes metabolizing DA such as monoamine oxidase B (MAO B) inhibitors (e.g., *selegiline*, *rasagiline*), or catechol-*O*-methyltransferase (COMT) inhibitors (e.g., *entacapone*, *tolcapone*). Moreover, blockade of histamine H_3_ receptors (H_3_R) could increase the level of DA in the brain. Histamine H_3_R are presynaptic receptors mainly distributed in the central nervous system, especially in the region connected with memory and learning. As heteroreceptors, they are located at the endings of non-histaminergic neurons (e.g., DA, acetylcholine, noradrenaline, serotonin). Deactivation of these receptors leads to enhanced release of the proper neurotransmitters including DA. PD is neurodegenerative disorder with a complicated etiology, and it is currently believed that only the use of drugs acting on several biological targets simultaneously can be effective in its treatment [1]. Thus, the search for multi-target drugs has developed in the last few years [1,2]. This novel strategy in drug design and development focuses on a combination of classical targets (e.g., MAO B inhibition) with new targets e.g., adenosine A_2A_ receptor blockade [3] or histamine H_3_R inhibition [4,5,6,7]. The concept of dual target ligands (DTL) linking blockade of MAO B with inhibition of H_3_R emerged a few years ago when preliminary screening for inhibitory activity toward human MAO B (hMAO B) showed promising inhibition of this enzyme by H_3_R ligands: ciproxifan (IC_50_ = 2 µM; Figure 1) and DL77 (IC_50_ = 19 nM; Figure 1) [6,8]. Thereafter, designed molecules were created as hybrids combining elements responsible for interaction with H_3_R (piperidine propyloxyphenyl element), and an MAO B motif e.g., a propargylamine moiety (**1**&**2**; Figure 1). These compounds showed stronger affinity for human H_3_R (*h*H_3_R) than hMAO B inhibitory activity. By contrast, some analogues of DL77 synthesized by our group with 4-*tert*-pentylphenyl moiety showed higher inhibitory activity for hMAO B than affinity for *h*H_3_R, e.g., compound **3** (*h*H_3_R *K_i_* = 63 nM; hMAO B IC_50_= 4.5 nM; Figure 1) [6].

Recently, we described DTL with the 4-*tert*-butylphenyl scaffold as *h*H_3_R ligands and hMAO B inhibitors [7]. This study is a continuation of the previous work with further structural modification of the lead compound DL76 (dual target activity: *h*H_3_R *K_i_* = 38 nM, hMAO B IC_50_ = 48 nM [7]) (Figure 1). Three types of modifications were introduced in the lead structure (Figure 2). All compounds obtained were tested for affinity to *h*H_3_R stably expressed in CHO or HEK293 cells. The inhibitory activity against hMAO B was evaluated by fluorometric MAO assay. For the two most potent hMAO B inhibitors (**9** and **13**; Table 1), the modality of hMAO B inhibition was assessed as well as an ability to cross the blood–brain barrier by using artificial membrane permeation assay (PAMPA). Next, the compound **13** was selected for further in vivo tests. The assessment concerned the effects of **13** on the feeding behavior of rats after its repeated peripheral injections and the influence on MAO A and B, and histamine *N*-methyltransferase (HNMT) activities, as well as cerebral catecholamine and serotonin concentrations. 

## 2. Materials and Methods

### 2.1. Synthesis of Compounds

This study is a continuation of the previous work and includes further structural modification of the lead compound DL76 (dual target activity: *h*H_3_R *K_i_* = 38 nM, hMAO B IC_50_ = 48 nM [7]) (Figure 1). The designed structural modification included: (a) change of the piperidine ring for other amines (cyclic or dialkyl), (b) change of a position or the kind of *tert*-butyl substituent, and (c) change of an ether linker for a carbamate linker.

All designed modifications are shown in Figure 2 and structures are collected in Appendix A.

Reagents and solvents were obtained from commercial suppliers and used without further purification. Reactions were conducted in the air atmosphere and monitored by thin layer chromatography (Merck silica gel 60 F254 plates). The spot visualization was achieved with UV lamp and Dragendorff’s reagent (solvent system: methylene chloride: methanol 9:1 or 1:1). Purity of compounds was confirmed by NMR spectra (^1^H and ^13^C) in DMSO-d_6_ using Mercury 300 MHz PFG spectrometer (Varian, Palo Alto, CA, USA) or FT-NMR 500 MHz spectrometer (Joel Ltd., Akishima, Tokyo, Japan). The chemical shifts (*δ*) are reported in relation to tetramethylsilane (TMS) and the coupling constants (J) are expressed in Hz. The multiplicity of each peak is reported as: s, singlet; d, doublet; t, triplet; q, quartet; quin, quintet; m, multiplet; br, broad; def, deformed. Mass spectra (LC/MS) were performed on Waters TQ Detector Mass Spectrometer (Water Corporation, Milford, CT, USA). Retention times (t_R_) are given in minutes. UPLC/MS analysis confirmed purity of compounds ≥97% (except **25**: 93%). The elemental analysis (C, H, N) for compounds (**4**–**7**; **9**; **11**; **13**–**16**; **19**; **24**,**25**; **27**–**29**) was performed on Vario EL III Elemental Analyser (Hanau, Germany) and results agreed within 0.5% of the theoretical value.

4-*tert-*Butylphenoxypropyl bromide (Ia) (CAS3245-63-4) was synthesized as described previously [7]. Other phenoxypropylbromides (Ib-Il) were obtained as Ia and all of them (except Ij) are reported in Chemical Abstract Database: 1-(4-(3-bromopropoxy)phenyl)ethan-1-one (Ib): CAS65623-98-5; 1-(3-bromopropoxy)-4-isopropylbenzene (Ic): CAS204979-21-5; 1-(3-bromopropoxy)-4-ethylbenzene (Id): CAS130402-63-0; 1-(3-bromopropoxy)-4-methylbenzene (Ie): CAS16929-24-1; 1-(3-bromopropoxy)-4-flurobenzene (If): CAS1129-78-8; 1-(3-bromopropoxy)-4-chlorobenzene (Ig): CAS27983-04-6; 2-*tert-*butylphenoxypropyl bromide (Ih) CAS414900-40-6; 3-*tert-*butylphenoxypropyl bromide (Ii) CAS1094702-92-7; 2-(3-bromopropoxy)-1-(*tert*-butyl)-3-methylbenzene (Ik) CAS1094273-48-9;1-(3-bromopropoxy)-2-(*tert-*butyl)-4-methylbenzene (Il) CAS1092405-68-6.


*Synthesis of below compounds were previously reported:*


*1-(3-(4-(tert-Butyl)phenoxy)propyl)-2-methylpiperidine hydrogen oxalate* (**10**) [7]

*1-(3-(4-(tert-Butyl)phenoxy)propyl)-pyrrolidine hydrogen oxalate* (**12**) [7]

*3-(Piperidin-1-yl)propyl tert-butylcarbamate hydrogen oxalate* (**30**) [10]

*3-(Piperidin-1-yl)propyl (2,4,4-trimethylpentan-2-yl)carbamate hydrogen oxalate* (**31**) [10]

*3-(Piperidin-1-yl)propyl (3,3-dimethylbutyl)carbamate hydrogen oxalate* (**32**) [10]

*General synthetic preparation of compounds***4**–**9**, **11**, **13**–**28**.

To a proper phenoxypropyl bromide (5 mmol) in acetonitrile (25 mL) and in the presence of K_2_CO_3_ (6 mmol) with the catalytic amount of KI was added a proper amine (5 mmol) and the solution was refluxed from 10 to 72 h. Next, a solid was filtered off and the oily residue was purified by flash chromatography (CH_2_Cl_2_:CH_3_OH, 50:50). The final product was transformed into oxalic acid salt in absolute C_2_H_5_OH and precipitated (C_2_H_5_)_2_O, or the solid was crystallized from C_2_H_5_OH. 

*3-(4-(tert-Butyl)phenoxy)-N,N-dimethylpropan-1-amine hydrogen oxalate* (**4**)

The title compound was prepared using dimethylamine (0.23 g, 5 mmol) and 4-*tert*-butylphenoxy propyl bromide (1.36 g, 5 mmol). Yield 10%, m.p. 140 dec °C, C_15_H_25_NO × C_2_H_2_O_4_ × 0.50 H_2_O (MW = 334.42). ^1^H NMR (500 MHz, DMSO-d_6_) δ: 7.25 (d, *J* = 8.6 Hz, 2H), 6.81 (d, *J* = 8.6 Hz, 2H), 3.96 (t, *J* = 6.3 Hz, 2H), 3.16—3.04 (m, 2H), 2.71 (s, 6H), 2.13–1.97 (m, 2H), 1.21 (s, 9H). ^13^C NMR (126 MHz, DMSO-d_6_) δ: 165.2, 156.6, 143.5, 126.6, 114.5, 65.3, 54.7, 42.8, 34.3, 31.9, 24.6. LC-MS: purity 100% t_R_ = 5.01, (ESI) *m/z* [M + H]^+^ 236.06. Analysis calculated for C_17_H_28_NO_5.5_: C, 61.05; H, 8.37; N, 4.19%. Found: C, 61.22; H, 8.61; N, 4.10%.

*3-(4-(tert-Butyl)phenoxy)-N-ethyl-N-methylpropan-1-amine hydrogen oxalate* (**5**)

The title compound was prepared using *N*-methylethamine (0.30 g, 5 mmol) and 4-*tert*-butylphenoxy propyl bromide (1.36 g, 5 mmol). Yield 7%, m.p. 131 dec °C, C_16_H_27_NO × C_2_H_2_O_4_× 0.25H_2_O (MW = 343.94). ^1^H NMR (500 MHz, DMSO-d_6_) δ: 7.25 (d, *J* = 8.59 Hz, 2H), 6.81 (d, *J* = 8.59 Hz, 2H), 3.97 (t, *J* = 5.73 Hz, 2H), 2.96–3.21 (m, 4H), 2.68 (s, 3H), 2.11–1.96 (m, 2H), 1.21 (s, 9H), 1.16 (t, *J* = 7.45 Hz, 3H). ^13^C NMR (126 MHz, DMSO-d_6_) δ: 165.3, 156.6, 143.5, 126.6, 114.5, 65.4, 52.3, 50.4, 39.2, 34.3, 31.9, 24.2, 9.5. LC-MS: purity 100% t_R_ = 5.21, (ESI) *m/z* [M + H]^+^ 250.15. Analysis calculated for C_18_H_29.5_NO_5.25_: C, 62.85; H, 8.40; N, 4.07%. Found: C, 62.80; H, 8.77; N, 4.00%.

*3-(4-(tert-Butyl)phenoxy)-N-isopropyl-N-methylpropan-1-amine hydrogen oxalate* (**6**)

The title compound was prepared using *N*-methylpropan-2-amine (0.37 g, 5 mmol) and 4-*tert*-butylphenoxy propyl bromide (1.36 g, 5 mmol). Yield 3%, m.p. 128 dec °C, C_17_H_29_NO × C_2_H_2_O_4_ ×0.5H_2_O (MW = 362.47). ^1^H NMR (500 MHz, DMSO-d_6_) δ: 7.25 (d, *J* = 8.59 Hz, 2H), 6.82 (d, *J* = 9.17 Hz, 2H), 3.97 (t, *J* = 6.01 Hz, 2H), 3.42–3.55 (m, 1H), 2.97–3.19 (m, 2H), 2.54–2.65 (m, 3H), 2.05 (dd, *J* = 6.59, 8.88 Hz, 2H), 1.21 (s, 9H), 1.17 (d, *J* = 6.30 Hz, 6H). ^13^C NMR (126 MHz, DMSO-d_6_) δ: 165.2, 156.6, 143.5, 126.6, 114.5, 65.3, 56.3, 50.0, 35.3, 34.3, 31.9, 24.7, 16.5. LC-MS: purity 100% t_R_ = 5.37, (ESI) *m/z* [M + H]^+^ 264.10. Analysis calculated for C_19_H_32_NO_5.5_: C, 62.95; H, 8.83; N, 3.87%. Found: C, 63.10; H, 8.99; N, 3.87%.

*3-(4-(tert-Butyl)phenoxy)-N,N-diethylpropan-1-amine hydrogen oxalate* (**7**)

The title compound was prepared using diethylamine (0.37 g, 5 mmol) and 4-*tert*-butylphenoxy propyl bromide (1.36 g, 5 mmol). Yield 10%, m.p. 105–108 °C, C_17_H_29_NO x C_2_H_2_O_4_× 1.5H_2_O (MW = 380.49). ^1^H NMR (300 MHz, DMSO-d_6_) δ: 7.28 (d, *J* = 8.79 Hz, 2H), 6.84 (d, *J* = 8.79 Hz, 2H), 4.00 (t, *J* = 5.86 Hz, 2H), 3.14 (quin, *J* = 7.33 Hz, 6H), 1.95–2.13 (m, 2H), 1.04–1.34 (m, 15H). ^13^C NMR (126 MHz, DMSO-d_6_) δ: 164.2, 156.5, 143.5, 126.6, 114.5, 65.2, 48.4, 46.7, 34.3, 31.9, 23.7, 9.0. LC-MS: purity 100% t_R_ = 5.42, (ESI) *m/z* [M + H]^+^ 264.24. Analysis calculated for C_19_H_34_NO_6.5_: C, 59.97; H, 8.94; N, 3.68%. Found: C, 60.05; H, 8.52; N, 3.45%.

*3-(4-(tert-Butyl)phenoxy)-N,N-diisopropylpropan-1-amine hydrogen chloride* (**8**)

The title compound was prepared using diizopropylamine (0.51 g, 5 mmol) and 4-*tert*-butylphenoxy propyl bromide (1.36 g, 5 mmol). Yield 5%, m.p. 193–195 °C, C_19_H_33_NO x HCl (MW = 327.92). ^1^H NMR (500 MHz, DMSO-d_6_) δ: 10.01–10.36 (m, 1H), 7.26 (d, *J* = 8.59 Hz, 2H), 6.82 (d, *J* = 8.59 Hz, 2H), 3.97 (t, *J* = 6.01 Hz, 2H), 3.29–3.36 (m, 8H), 3.07–3.20 (m, 2H), 2.73 (s, 6H), 2.00–2.14 (m, 2H), 1.21 (s, 9H). ^13^C NMR (126 MHz, DMSO-d_6_) δ: 156.5, 143.5, 126.6, 114.5, 65.3, 54.5, 42.5, 34.3, 31.9, 24.4. LC-MS: purity 100% t_R_ = 5.04, (ESI) *m/z* [M + H]^+^ 236.32.

*3-(4-(tert-Butyl)phenoxy)-N,N-dipropylpropan-1-amine hydrogen oxalate* (**9**)

The title compound was prepared using dipropylamine (0.51 g, 5 mmol) and 4-*tert*-butylphenoxy propyl bromide (1.36 g, 5 mmol). Yield 25%, m.p. 139–142 °C, C_19_H_33_NO x C_2_H_2_O_4_ (MW = 381.51). ^1^H NMR (300 MHz, DMSO-d_6_) δ: 7.28 (d, *J* = 8.79 Hz, 2H), 6.83 (d, *J* = 8.79 Hz, 2H), 4.00 (t, *J* = 5.86 Hz, 2H), 3.05–3.23 (m, 2H), 2.79–3.05 (m, 4H), 1.95–2.13 (m, 2H), 1.49–1.76 (m, 4H), 1.23 (s, 9H), 0.88 (t, *J* = 14.70 Hz, 6H). ^13^C NMR (126 MHz, DMSO-d_6_) δ: 165.1, 156.5, 143.5, 126.6, 114.5, 65.3, 54.0, 49.5, 34.3, 31.9, 23.8, 17.2, 11.5. LC-MS: purity 100% t_R_ = 5.89, (ESI) *m/z* [M + H]^+^ 292.21. Analysis calculated for C_21_H_35_NO_5_: C, 66.11; H, 9.25; N, 3.67%. Found: C, 65.80; H, 9.26; N, 3.60%.

*1-(3-(4-(tert-Butyl)phenoxy)propyl)-2,6-dimethylpiperidine hydrogen chloride* (**11**)

The title compound was prepared using 2,6-dimethylpiperidine (0.57 g, 5 mmol) and 4-*tert*-butylphenoxy propyl bromide (1.36 g, 5 mmol). Yield 10%, m.p. 194–196 dec °C,C_20_H_33_NO × HCl × 0.25H_2_O (MW = 344.46). ^1^H NMR (DMSO-d_6_, 500 MHz) δ: 9.7 (br s, 1H), 7.24 (d, *J*=8.9 Hz, 2H), 6.79–6.85 (m, 2H), 4.00 (t, 2H, *J*=5.4 Hz), 3.23 (br s, 4H,), 1.90–2.09 (m, 2H), 1.78 (br d, 2H, *J*=12.9 Hz), 1.39–1.66 (m, 4H), 1.20–1.29 (m, 6H), 1.19 (s, 9H). ^13^C NMR (126 MHz, DMSO-d_6_) δ: 156.4, 143.6, 126.7, 126.6, 114.5, 114.4, 64.9, 60.7, 58.3, 44.7, 34.3, 32.0, 31.8, 25.3, 22.5, 21.3, 18.1, 17.3. LC-MS: purity 100% t_R_ = 5.83, (ESI) *m/z* [M + H]^+^ 304.37. Analysis calculated for C_25_H_34.5_NO_1.25_Cl: C, 69.67; H, 10.02; N, 4.07%. Found: C, 69.93; H, 10.49; N, 3.94%.

*1-(3-(4-(tert-Butyl)phenoxy)propyl)-2-methylpyrrolidine hydrogen oxalate* (**13**)

The title compound was prepared using 2-methylpyrrolidine (0.43 g, 5 mmol) and 4-*tert*-butylphenoxy propyl bromide (1.36 g, 5 mmol). Yield 10%, m.p. 109–111 °C, C_18_H_29_NO × C_2_H_2_O_4_ (MW = 365.47). ^1^H NMR (500 MHz, DMSO-d_6_) δ: 7.25 (d, *J* = 8.59 Hz, 2H), 6.82 (d, *J* = 8.88 Hz, 2H), 3.93–4.03 (m, 2H), 3.48–3.62 (m, 1H), 3.34 (d, *J* = 7.73 Hz, 2H), 2.92–3.14 (m, 2H), 2.00–2.17 (m, 3H), 1.81–1.96 (m, 2H), 1.52–1.66 (m, 1H), 1.28 (d, *J* = 6.30 Hz, 3H), 1.20 (s, 9H). ^13^C NMR (126 MHz, DMSO-d_6_) δ: 165.4, 156.6, 143.4, 126.6, 114.5, 65.4, 63.2, 52.7, 49.6, 34.3, 31.9, 31.4, 25.7, 21.4, 15.7. LC-MS: purity 100% t_R_ = 5.46, (ESI) *m/z* [M + H]^+^ 276.32. Analysis calculated for C_20_H_31_NO_5_: C, 65.73; H, 8.55; N, 3.83%. Found: C, 65.53; H, 8.82; N, 3.71%.

*4-(3-(4-(tert-Butyl)phenoxy)propyl)morpholine hydrogen oxalate* (**14**)

The title compound was prepared using morpholine (0.44 g, 5 mmol) and 4-*tert*-butylphenoxy propyl bromide (1.36 g, 5 mmol). Yield 19%, m.p. 188–192 °C, C_17_H_27_NO_2_× C_2_H_2_O_4_ (MW = 367.44). ^1^H NMR (300 MHz, DMSO-d_6_) δ: 7.27 (d, *J* = 8.79 Hz, 2H), 6.83 (d, *J* = 8.79 Hz, 2H), 3.97 (t, *J* = 6.15 Hz, 2H), 3.72 (br. s., 4H), 2.92 (br. s., 6H), 1.88–2.09 (m, 2H), 1.03–1.37 (m, 9H). ^13^C NMR (126 MHz, DMSO-d_6_) δ: 164.6, 156.6, 143.4, 126.6, 114.5, 65.6, 64.7, 54.4, 52.2, 34.3, 31.9, 24.4. LC-MS: purity 97% t_R_ = 4.97, (ESI) *m/z* [M + H]^+^ 278.19. Anal calculated for C_19_H_29_NO_6.5_: C, 62.11; H, 7.96; N, 3.81%. Found: C, 61.96; H, 8.41; N, 3.69%.

*1-(3-(4-(tert-Butyl)phenoxy)propyl)-4-methylpiperazine hydrogen oxalate* (**15**)

The title compound was prepared using 4-methylpiperazine (0.50 g, 5 mmol) and 4-*tert*-butylphenoxy propyl bromide (1.36 g, 5 mmol). Yield 19%, m.p. 172 dec °C, C_18_H_30_N_2_O × 2C_2_H_2_O_4_ (MW = 470.52). ^1^H NMR (300 MHz, DMSO-d_6_) δ: 7.26 (d, *J* = 8.79 Hz, 2H), 6.81 (d, *J* = 8.79 Hz, 2H), 3.95 (t, *J* = 6.15 Hz, 2H), 3.02 (br. s., 4H), 2.53–2.88 (m, 9H), 1.88 (quin, *J* = 6.45 Hz, 2H), 1.22 (s, 9H). ^13^C NMR (126 MHz, DMSO-d_6_) δ: 163.9, 156.8, 143.2, 126.6, 114.4, 65.8, 54.0, 52.7, 50.2, 43.2, 34.3, 31.9, 26.0. LC-MS: purity 100% t_R_ = 4.39, (ESI) *m/z* [M + H]^+^ 291.22. Analysis calculated for C_22_H_34_N_2_O_9_: C, 56.16; H, 7.28; N, 5.95%. Found: C, 56.19; H, 7.43; N, 6.04%.

*1-(3-(4-(tert-Butyl)phenoxy)propyl)-4-phenylpiperazine hydrogen oxalate* (**16**)

The title compound was prepared using 4-phenylpiperazine (0.81 g, 5 mmol) and 4-*tert*-butylphenoxy propyl bromide (1.36 g, 5 mmol). Yield 15%, m.p. 164–166 °C, C_23_H_32_N_2_O × C_2_H_2_O_4_ (MW = 442.56). ^1^H NMR (300 MHz, DMSO-d_6_) δ: 7.18–7.33 (m, 4H), 6.97 (d, *J* = 8.21 Hz, 2H), 6.77–6.91 (m, 3H), 4.00 (t, *J* = 5.86 Hz, 2H), 3.40 (br. s., 4H), 3.14 (br. s., 6H), 1.97–2.17 (m, 2H), 1.23 (s, 9H). ^13^C NMR (126 MHz, DMSO-d_6_) δ: 164.6, 156.6, 150.5, 143.4, 129.6, 126.6, 120.2, 116.3, 114.5, 65.6, 54.0, 51.8, 46.7, 34.3, 31.9, 24.7. LC-MS: purity 100% t_R_ = 6.01, (ESI) *m/z* [M + H]^+^ 353.22. Analysis calculated for C_25_H_34_N_2_O_5_: C, 67.85; H, 7.68; N, 6.33%. Found: C, 67.77; H, 8.12; N, 6.28%.

(*Z)-1-(3-(4-(tert-Butyl)phenoxy)propyl)-4-(3-phenylallyl)piperazine hydrogen oxalate* (**17**)

The title compound was prepared using 1-cinnamylpiperazine (1.01 g, 5 mmol) and 4-*tert*-butylphenoxy propyl bromide (1.36 g, 5 mmol). Yield 19%, m.p. 217–219 °C, C_26_H_36_N_2_O ×2C_2_H_2_O_4_ (MW = 572.67). ^1^H NMR (300 MHz, DMSO-d_6_) δ: 7.39–7.51 (m, 2H), 7.30–7.39 (m, 2H), 7.21–7.30 (m, 3H), 6.82 (d, *J* = 8.79 Hz, 2H), 6.66 (d, *J* = 15.82 Hz, 1H), 6.20–6.38 (m, 1H), 3.97 (t, *J* = 5.86 Hz, 2H), 3.44 (d, *J* = 6.45 Hz, 2H), 2.62–3.17 (m, 10H), 1.88–2.07 (m, 2H), 1.22 (s, 9H). ^13^C NMR (126 MHz, DMSO-d_6_) δ: 165.0, 163.7, 156.7, 143.3, 136.6, 135.3, 129.2, 128.5, 127.0, 126.6, 114.4, 65.7, 59.0, 53.9, 51.2, 50.6, 34.3, 31.9, 25.3. LC-MS: purity 100% t_R_ = 6.17, (ESI) *m/z* [M + H]^+^ 393.23.

*1-(4-(3-(Piperidin-1-yl)propoxy)phenyl)ethan-1-one hydrogen oxalate* (**18**)

The title compound was prepared using piperidine (0.43 g, 5 mmol) and 1-(4-(3-bromopropoxy)phenyl)ethan-1-one (1.29 g, 5 mmol). Yield 13%, m.p. 157–159 °C, C_16_H_23_NO_2_× C_2_H_2_O_4_ (MW = 351.44). ^1^H NMR (500 MHz, DMSO-d_6_) δ: 7.89 (d, *J* = 8.88 Hz, 2H), 7.00 (d, *J* = 8.88 Hz, 2H), 4.09 (t, *J* = 6.01 Hz, 2H), 2.86–3.25 (m, 6H), 2.48 (s, 3H), 2.04–2.16 (m, 2H), 1.60–1.76 (m, 4H), 1.49 (br. s., 2H). ^13^C NMR (126 MHz, DMSO-d_6_) δ: 196.9, 165.2, 162.6, 131.0, 130.6, 114.8, 65.9, 53.8, 52.7, 27.0, 23.9, 23.2, 22.1. LC/MS: purity: 100%, t_R_ = 2.95, (ESI) *m/z* [M + H]^+^ 262.18.

*1-(3-(4-Isopropylphenoxy)propyl)piperidine hydrogen oxalate* (**19**)

The title compound was prepared using piperidine (0.43 g, 5 mmol) and 4-izopropylphenoxy propyl bromide (1.29 g, 5 mmol). Yield 23%, m.p. 112–115 °C, C_17_H_27_NO × C_2_H_2_O_4_ (MW = 351.44). ^1^H NMR (300 MHz, DMSO-d_6_) δ: 7.13 (d, *J* = 8.79 Hz, 2H), 6.83 (d, *J* = 8.21 Hz, 2H), 3.97 (t, *J* = 5.86 Hz, 2H), 3.09 (t, *J* = 7.60 Hz, 6H), 2.81 (spt, *J* = 6.80 Hz, 1H), 1.96–2.17 (m, 2H), 1.60–1.82 (m, 4H), 1.50 (br. s., 2H), 1.14 (d, *J* = 6.45 Hz, 6H). ^13^C NMR (126 MHz, DMSO-d_6_) δ: 165.2, 156.9, 141.2, 127.7, 114.8, 65.5, 54.1, 52.6, 33.1, 24.7, 24.1, 23.2, 22.0. LC-MS: purity 100% t_R_ = 5.03, (ESI) *m/z* [M + H]^+^ 262.24. Analysis calculated for C_19_H_29_NO_5_: C, 64.93; H, 8.32; N, 3.99%. Found: C, 64.52; H, 8.49; N, 3.89%. 

*1-(3-(4-Ethylphenoxy)propyl)piperidine hydrogen oxalate* (**20**)

The title compound was prepared using piperidine (0.43 g, 5 mmol) and 4-ethylphenoxy propyl bromide (1.22 g, 5 mmol). Yield 37%, m.p. 144–146 °C, C_16_H_25_NO × C_2_H_2_O_4_ (MW = 337.48). ^1^H NMR (300 MHz, DMSO-d_6_) δ: 7.10 (d, *J* = 8.79 Hz, 2H), 6.82 (d, *J* = 8.79 Hz, 2H), 3.97 (t, *J* = 5.86 Hz, 2H), 2.79–3.42 (m, 6H), 2.52 (s, 1H), 1.95–2.16 (m, 2H), 1.59–1.83 (m, 4H), 1.51 (br. s., 2H), 1.12 (t, *J* = 7.62 Hz, 3H). ^13^C NMR (126 MHz, DMSO-d_6_) δ: 165.2, 156.8, 136.5, 129.2, 114.9, 65.5, 54.1, 52.7, 27.8, 24.1, 23.2, 22.0, 16.5. LC/MS: purity: 100%, t_R_ = 4.52, (ESI) *m/z* [M + H]^+^ 248.22.

*1-(3-(p-Tolyloxy)propyl)piperidine hydrogen oxalate* (**21**)

The title compound was prepared using piperidine (0.43 g, 5 mmol) and 4-methylphenoxy propyl bromide (1.14 g, 5 mmol). Yield 21%, m.p. 160–162 °C, C_15_H_23_NO × C_2_H_2_O_4_ (MW = 323.43). ^1^H NMR (300 MHz, DMSO-d_6_) δ: 7.07 (d, *J* = 8.21 Hz, 2H), 6.80 (d, *J* = 8.21 Hz, 2H), 3.96 (t, *J* = 5.86 Hz, 2H), 2.93–3.24 (m, 6H), 2.21 (s, 3H), 2.00–2.13 (m, 2H), 1.63–1.79 (m, 4H), 1.50 (br. s., 2H). ^13^C NMR (126 MHz, DMSO-d_6_) δ: 165.3, 156.7, 130.4, 129.9, 114.8, 65.5, 54.0, 52.6, 24.1, 23.1, 22.0, 20.6. LC/MS: purity: 98.9%, t_R_=3.98, (ESI) m/z [M + H]^+^ 234.20. 

*1-(3-(4-Fluorophenoxy)propyl)piperidine hydrogen oxalate* (**22**)

The title compound was prepared using piperidine (0.43 g, 5 mmol) and 4-fluorophenoxy propyl bromide (1.17 g, 5 mmol). Yield 15%, m.p. 132–136 °C, C_14_H_19_NOF × C_2_H_2_O_4_ (MW = 327.38). ^1^H NMR (300 MHz, DMSO-d_6_) δ: 7.10 (t, *J* = 8.79 Hz, 2H), 6.89–6.98 (m, 2H), 3.99 (t, *J* = 5.86 Hz, 2H), 2.89–3.25 (m, 6H), 1.98–2.16 (m, 2H), 1.62–1.80 (m, 4H), 1.50 (br. s., 2H). ^13^C NMR (126 MHz, DMSO-d_6_) δ: 165.3, 158.0, 156.2, 155.1, 155.1, 116.5, 116.3, 116.3, 116.2, 66.1, 54.0, 52.6, 24.0, 23.1, 22.0. LC/MS: purity: 98.7%, t_R_ = 3.57, (ESI) m/z [M + H]^+^ 238.12.

*1-(3-(4-chlorophenoxy)propyl)piperidine hydrogen oxalate* (**23**)

The title compound was prepared using piperidine (0.43 g, 5 mmol) and 4-chlorophenoxy propyl bromide (1.25 g, 5 mmol). Yield 54%, m.p. 158–160 °C, C_14_H_20_NOCl × C_2_H_2_O_4_ (MW = 348.73). ^1^H NMR (300 MHz, DMSO-d_6_) δ: 7.31 (d, *J* = 8.79 Hz, 2H), 6.94 (d, *J* = 8.79 Hz, 2H), 4.00 (t, *J* = 5.86 Hz, 2H), 2.87–3.29 (m, 6H), 2.00–2.17 (m, 2H), 1.59–1.81 (m, 4H), 1.50 (br. s., 2H). ^13^C NMR (126 MHz, DMSO-d_6_) δ: 165.3, 157.6, 129.8, 125.0, 116.8, 65.9, 53.9, 52.6, 23.9, 23.1, 22.0. LC/MS: purity: 100%, t_R_ = 4.11, (ESI) m/z [M + H]^+^ 254.13. 

*1-(3-(2-(tert-Butyl)phenoxy)propyl)piperidine hydrogen oxalate* (**24**)

The title compound was prepared using piperidine (0.43 g, 5 mmol) and 2-*tert*-butylphenoxy propyl bromide (1.36 g, 5 mmol). Yield 8%, m.p. 184 dec °C, C_18_H_29_NO x C_2_H_2_O_4_× 0.25H_2_O (MW = 369.98). ^1^H NMR (300 MHz, DMSO-d_6_) δ: 7.11–7.27 (m, 2H), 6.93 (d, *J* = 7.62 Hz, 1H), 6.86 (t, *J* = 7.30 Hz, 1H), 4.03 (t, *J* = 6.15 Hz, 2H), 2.95–3.28 (m, 6H), 2.08–2.24 (m, 2H), 1.64–1.82 (m, 4H), 1.52 (d, *J* = 3.52 Hz, 2H), 1.32 (s, 9H). ^13^C NMR (126 MHz, DMSO-d_6_) δ: 165.2, 157.5, 137.6, 127.7, 126.8, 120.9, 112.9, 65.5, 54.4, 52.8, 34.9, 30.3, 24.4, 23.3, 22.1. LC-MS: purity 98.6% t_R_ = 5.24, (ESI) *m/z* [M + H]^+^ 276.20. Analysis calculated for C_20_H_31.5_NO_5.25_Cl: C, 64.87; H, 8.51; N, 3.79%. Found: C, 65.08; H, 8.49; N, 3.76%.

*1-(3-(3-(tert-Butyl)phenoxy)propyl)piperidine hydrogen chloride* (**25**)

The title compound was prepared using piperidine (0.43 g, 5 mmol) and 3-*tert*-butylphenoxy propyl bromide (1.36 g, 5 mmol). Yield 4%, m.p. 145–148 °C, C_18_H_29_NO x HCl × 0.25H_2_O (MW = 316.40). ^1^H NMR (300 MHz, DMSO-d_6_) δ: 10.12 (br. s., 1H), 7.20 (t, *J* = 7.91 Hz, 1H), 6.96 (d, *J* = 8.21 Hz, 1H), 6.88 (s, 1H), 6.74 (dd, *J* = 2.05, 7.91 Hz, 1H), 4.02 (t, *J* = 5.86 Hz, 2H), 3.43 (d, *J* = 11.72 Hz, 2H), 3.15 (br. s., 2H), 2.85 (br. s., 2H), 2.04–2.22 (m, 2H), 1.60–1.89 (m, 5H), 1.12–1.50 (m, 10H). ^13^C NMR (126 MHz, DMSO-d_6_) δ: 158.6, 152.9, 129.6, 118.3, 112.6, 111.3, 65.3, 54.0, 52.5, 35.0, 31.6, 23.9, 22.9, 21.9. LC-MS: purity 93.4% t_R_ = 5.39, (ESI) *m/z* [M + H]^+^ 276.26. Analysis calculated for C_18_H_30.5_NO_1.25_Cl: C, 68.27; H, 9.64; N, 4.45%. Found: C, 68.40; H, 9.74; N, 4.30%.

*1-(3-(2-(tert-Butyl)-6-methylphenoxy)propyl)piperidine hydrogen oxalate* (**26**)

The title compound was prepared using piperidine (0.43 g, 5 mmol) and 2-*tert*-butyl-6-methylphenoxy propyl bromide (1.43 g, 5 mmol). Yield 3%, m.p. 134–135 °C, C_19_H_31_NO × C_2_H_2_O_4_ (MW = 379.50). ^1^H NMR (300 MHz, DMSO-d_6_) δ: 7.11 (d, *J* = 6.45 Hz, 1H), 7.05 (d, *J* = 7.62 Hz, 1H), 6.94 (def t, 1H), 3.81 (br. s., 2H), 3.23 (br. s., 3H), 2.68–3.20 (m, 3H), 2.23 (s, 3H), 2.16 (br. s., 2H), 1.73 (br. s., 4H), 1.53 (br. s., 2H), 1.32 (s, 9H). ^13^C NMR (126 MHz, DMSO-d_6_) δ: 164.4, 156.5, 142.4, 131.5, 130.4, 125.3, 124.0, 69.4, 53.9, 52.7, 35.2, 31.6, 24.9, 23.2, 21.9, 17.5. LC-MS: purity 100% t_R_ = 5.52, (ESI) *m/z* [M + H]^+^ 290.29.

*1-(3-(2-(tert-Butyl)-5-methylphenoxy)propyl)piperidine hydrogen oxalate* (**27**)

The title compound was prepared using piperidine (0.43 g, 5 mmol) and 2-*tert*-butyl-5-methylphenoxy propyl bromide (1.43 g, 5 mmol). Yield 49%, m.p. 199 dec °C, C_19_H_31_NO × C_2_H_2_O_4_ (MW = 379.50). ^1^H NMR (300 MHz, DMSO-d_6_) δ: 7.06 (d, *J* = 7.62 Hz, 1H), 6.75 (s, 1H), 6.66 (d, *J* = 7.62 Hz, 1H), 4.01 (t, *J* = 5.86 Hz, 2H), 2.78–3.29 (m, 6H), 2.07–2.29 (m, 5H), 1.71 (br. s., 4H), 1.51 (br. s., 2H), 1.14–1.39 (m, 9H). LC-MS: purity 99.3% t_R_ = 5.64, (ESI) *m/z* [M + H]^+^ 290.29. Analysis calculated for C_21_H_33_NO_5_: C, 66.46; H, 8.77; N, 3.69%. Found: C, 66.18; H, 8.98; N, 3.61%.

*1-(3-(2-(tert-Butyl)-4-methylphenoxy)propyl)piperidine hydrogen oxalate* (**28**)

The title compound was prepared using piperidine (0.43 g, 5 mmol) and 2-*tert*-butyl-4-methylphenoxy propyl bromide (1.43 g, 5 mmol). Yield 33%, m.p. 182–184 °C, C_19_H_31_NO × C_2_H_2_O_4_ (MW = 379.50). ^1^H NMR (300 MHz, DMSO-d_6_) δ: 7.00 (d, *J* = 1.76 Hz, 1H), 6.94 (d, *J* = 8.21 Hz, 1H), 6.78–6.87 (m, 1H), 3.98 (t, *J* = 6.15 Hz, 2H), 2.91–3.26 (m, 6H), 2.04–2.28 (m, 5H), 1.70 (d, *J* = 4.69 Hz, 4H), 1.51 (br. s., 2H), 1.31 (s, 9H). ^13^C NMR (126 MHz, DMSO-d_6_) δ: 165.1, 155.4, 137.3, 129.2, 127.7, 127.6, 112.9, 65.6, 54.4, 52.8, 34.8, 30.4, 24.5, 23.3, 22.1, 21.0. LC-MS: purity 97.7% t_R_ = 5.74, (ESI) *m/z* [M + H]^+^ 290.29. Analysis calculated for C_21_H_33_NO_5_: C, 66.46; H, 8.77; N, 3.69%. Found: C, 66.08; H, 9.01; N, 3.72%.


*Synthesis of Compound*
**29**



*3-(Piperidin-1-yl)propyl (4-(tert-butyl)phenyl)carbamate hydrogen oxalate (*
**29**
*)*


To 1-(*tert*-butyl)-4-isocyanatobenzene (2.5 mmol, 0.438 g) dissolved in 20 mL of CH_3_CN (HPLC purity) 3-(piperidin-1-yl)propan-1-ol [10] (2.5 mmol, 0.361 g) in 10 mL of CH_3_CN was added and refluxed for 10 h. After that time, the solution was concentrated in vacuo and purified by flash chromatography (CH_2_Cl_2_:CH_3_OH; 80:20). The pure fractions were evaporated in vacuo and the final product was transformed into oxalic acid salt in absolute C_2_H_5_OH and precipitated (C_2_H_5_)_2_O. The solid was crystallized from C_2_H_5_OH. Yield 35%, m.p. 171 dec °C, C_19_H_30_N_2_O_2_× C_2_H_2_O_4_ (MW = 408.50). ^1^H NMR (500 MHz, DMSO-d_6_) δ: 9.54 (br. s., 1H), 7.29–7.45 (m, 2H), 7.25 (d, *J* = 8.59 Hz, 2H), 4.08 (t, *J* = 6.30 Hz, 2H), 2.82–3.33 (m, 6H), 1.88–2.11 (m, 2H), 1.68 (br. s., 4H), 1.48 (br. s., 2H), 1.20 (s, 9H). ^13^C NMR (126 MHz, DMSO-d_6_) δ: 165.1, 153.9, 145.3, 136.9, 125.9, 118.6, 62.1, 53.8, 52.6, 34.4, 31.7, 23.9, 23.1, 22.0.LC-MS: purity 100% t_R_ = 5.21, (ESI) *m/z* [M + H]^+^ 319.34. Analysis calculated for C_21_H_3_N_2_O_6_: C, 61.75; H, 7.90; N, 6.86%. Found: C, 61.27; H, 8.22; N, 6.65%.

### 2.2. Key Reagents (Cytotoxicity and In Vivo Pharmacology Studies)

Pitolisant, 3-(4-chlorophenyl)propyl 3-piperidinopropyl ether as hydrochloride salt, was synthesized in the Department of Technology and Biotechnology of Drugs, Jagiellonian University Medical College in Kraków (Kraków, Poland). 

Astrocyte medium, astrocyte growth supplement, fetal bovine serum, penicillin/streptomycin solution and poly-L-Lysine were from ScienCell Research Laboratories (Carlsbad, CA, USA). 

The 3-(4,5-dimethylthazol-2-yl)-2,5-diphenyltetrazolium bromide (MTT), clorgyline (*N*-methyl-*N*-propargyl-3-(2,4-dichlorophenoxy)propylamine hydrochloride), deprenyl (*R*-(-)-deprenyl hydrochloride) and dimethyl sulfoxide were purchased from Sigma-Aldrich, Inc. (St. Louis, MO, USA). 

The reagents for radioassays, i.e., β-phenylethylamine hydrochloride [ethyl-1-^14^C], hydroxytryptamine binoxalate (serotonin binoxalate), 5-[2-^14^C] and adenosyl-L-methionine, S-[methyl-^14^C] were obtained from American Radiolabeled Chemicals, Inc. (St. Louis, MO, USA).

### 2.3. In Vitro Biological Studies

#### 2.3.1. Histamine H_3_ Receptor Affinity

Affinity to *h*H_3_R stably expressed in CHO-K1 [6] or HEK293 [11] cells was evaluated in a radioligand binding assay as described previously. Briefly, 10 mM stock solutions of the test compounds in DMSO were prepared. Each compound was tested at eight concentrations ranging from 10^−5^ to 10^−12^ M (final concentration). All assays were carried out in duplicate. Crude membrane preparations were incubated with the tested compounds and [^3^H]*N*^α^-methylhistamine (radioligand; 2 nM; KD = 3.08 nM) in binding buffer (total volume 0.2 mL) for 60–90 min under continuous shaking. (R)(-)-α-methylhistamine (100 µM) [6] or pitolisant (10 µM) were used to define nonspecific binding. The radioactivity was counted in MicroBeta 2 [6] or MicroBeta Trilux [11] counter (PerkinElmer). Data were fitted to a one-site curve-fitting equation with Prism 6 (GraphPad Software, San Diego, CA, USA) and *K_i_* values were calculated from IC_50_ values (from at least three experiments performed in duplicates) according to the Cheng−Prusoff equation [12].

#### 2.3.2. Monoamine Oxidase B Inhibitory Activity

The precise method was described in [6]. First, compounds **4**–**32** were screened for hMAO B inhibitory activity at 1 μM concentration by the fluorometric method. *Para*-tyramine (200 μM) was used as a substrate for the enzyme and safinamide (1 μM) was used as a reference compound. IC_50_ values were evaluated for compounds that inhibited the enzyme by more than 50% of pargyline (10 μM) activity. Each experiment was performed in duplicate.

#### 2.3.3. Modality of Monoamine Oxidase B Inhibition 

Modality of hMAO B inhibition was evaluated for compounds **9** and **13** and the reference safinamide according to the method described previously [6,7]. Three concentrations of inhibitors, corresponding to their IC_20_, IC_50_ and IC_80_ values, were used. Each experiment was performed in triplicate. K_M_ and V_max_ values were calculated from Michaelis–Menten curves by nonlinear regression from the substrate. Lineweaver–Burk plots were calculated using linear regression in GraphPad Prism 6.07 (GraphPad Software, San Diego, CA, USA).

#### 2.3.4. Reversibility of Monoamine Oxidase B Inhibition

The reversibility of the MAO inhibition was tested as described in [6,7]. Compounds **9** and **13** were tested in the concentration corresponding to their IC_80_ along with reference reversible (safinamide) and irreversible (rasagiline) MAO B inhibitors. Two variants of experiment were performed. In the first variant, enzyme and inhibitors were added to the reaction mixture at the same time with lower concentration (10 µM) of the substrate (p-tyramine). After 22 min, the concentration of the substrate was increased to 200 µM and the signal was measured for 5 h. In the second variant, inhibitors and enzyme were preincubated for 30 min before the addition of the lower concentration of the substrate, with the next steps performing analogically to the first variant.

#### 2.3.5. Parallel Artificial Membrane Permeability 

To evaluate permeability, the pre-coated PAMPA Plate System (Gentest^TM^, Corning, Tewksbury, MA, USA) was used as we described previously [13]. Two compounds were selected for evaluation (**9** and **13**). Caffeine was used as the highly permeable reference. The concentrations of tested compounds were estimated by the LC/MS method on Waters TQ Detector Mass Spectrometer (Water Corporation, Milford, CT, USA) with the internal standard. The assay was conducted in triplicate. The permeability coefficients *Pe* (10^−6^cm/s) were calculated using the formula provided by the manufacturer.

#### 2.3.6. Evaluation of the Cytotoxicity of Compounds **9** and **13**

##### Cell Cultures

The studies were performed on a commercially-available astrocyte cell line isolated from human cerebral cortex (ScienCell Research Laboratories, San Diego, CA, USA; Cat no. 1800). The cells (passage 7–8) were seeded into a 96-well plate at an amount of 10,000 cells/well and kept in accordance with the ScienCell Research Laboratories’ protocol, i.e., in astrocyte medium supplemented with 2% fetal bovine serum, 10% astrocyte growth supplement and 1% penicillin/streptomycin solution, in an atmosphere with 5% CO_2_ at 37 °C. The cells were allowed to grow for 24 h and then treated with increasing concentrations of test compounds (0.01–0.25 mg/mL) for 24 and 72 h. Astrocytes in the medium on each plate (regardless of the factors tested) were used as a positive control. 

All procedures were performed in a laminar chamber ensuring sterile conditions.

##### MTT Cell Viability Test

The viability of the astrocyte cell line was determined calorimetrically using 3-(4,5-dimethylthazol-2-yl)-2,5-diphenyltetrazolium bromide (MTT; Sigma-Aldrich Chemical Co. Ltd., Saint Louis, MO, USA) as described earlier [14]. Cells placed into 96-well plates (10,000 cells/well), after 24 h of culture in standard conditions, were exposed to Compound **9**, Compound **13** or pitolisant. After the incubation time (24 or 72 h) with examined drugs, 50 μL MTT solution (1 mg/mL) was added to each well of the plate for another 4 h. The method is based on the reduction of a yellow tetrazolium salt (MTT) into purple formazan crystals by mitochondrial succinate-tetrazolium reductase system which is metabolically active in viable cells [15,16]. At the end of the experiment, the cells were treated with 100 μL dimethyl sulphoxide, which enabled the release of the reaction product. 

The absorbance was measured at 570 nm using a BioTek EL×800 microplate reader (BioTek, Winooski, Vermont, USA) and results were expressed as a percentage of the absorbance measured in control cells. The obtained values were plotted against different concentrations of each compound to calculate the viability inhibition concentration at 50% (IC_50_) using GraphPad Prism 6.07 (GraphPad Software, Inc., San Diego, CA, USA). The experiment was repeated in quadruplicate. 

For the cytotoxicity assessment in the MTT assays, each test drug was used at 8 concentrations (range: 0.01–0.25 mg/mL).

### 2.4. Animals and Pharmacological Treatment 

The compound **13** has been examined for its in vivo activity in rats. Male Wistar rats weighing 180–240 g at the beginning of the experiments were used for the study. Animals were individually housed in standard cages with liquid and food available ad libitum, under an artificial reversed 12-h light–dark cycle with light off at 7 a.m., temperature 21–22 °C and 60–65% humidity. 

Before the start of the experiments, the animals were habituated for 7 days to the conditions in the animal facility. Pharmacological treatments were carried out in the dark phase of the cycle. During the drug administration, the rats were kept in metabolic cages (Tecniplast, Italy). After an additional one day adaptation, there was a 3-day pre-treatment phase.

The compound **13** was given to intact Wistar rats to verify its impact on the metabolism of biogenic amines in the brain (cerebral amines and their metabolites concentrations as well as activities of metabolizing enzymes). Pitolisant (H_3_R antagonist/inverse agonist) was employed as a reference drug [17,18]. Both drugs (3 mg/kg body mass, dissolved in 0.9% NaCl) were given subcutaneously for 6 consecutive days. Control rats were injected with 200 μL of 0.9% NaCl. 

The volumes of consumed food and water, as well as urine excretion, were recorded daily and expressed in g or mL per 100 g of body mass or mL per 24 h, respectively. 

The final results are given as means with SEM calculated for each 24-h period, computed from 3-day (pre-treatment phase) or 6-day (pharmacological treatment) monitoring [19,20]. All experimental procedures were undertaken according to EU directives and local ethical regulations.

### 2.5. Sample Preparation and Biochemical Analyzes

Rats subjected to pharmacological treatment with the compound **13** and pitolisant (reference compound) were sacrificed by decapitation 2 h after the last drug administration. Tissues were collected and properly prepared for subsequent biochemical analyzes. The brain was quickly removed from the skull and the selected structures (hypothalamus, striatum, cerebral cortex) were dissected according to the method by Glowinsky and Iversen [21], immediately frozen in liquid nitrogen and kept at −80 °C until assayed. 

#### 2.5.1. HNMT and MAOs Activities

MAO A and MAO B activities were estimated in cerebral homogenates with radioassays using 5-[2-^14^C]-hydroxytryptamine binoxalate (final conc. 200 µM) and β-[ethyl-1-^14^C]-phenylethylamine hydrochloride (final conc. 20 µM), as well as specific inhibitors—clorgyline and deprenyl (final conc. 10^−9^ M), respectively [22,23]. 

Histamine *N*-methyltransferase activity was determined radioenzymatically according to Taylor and Snyder [24] by measurements of radioactive *N*-tele-methylhistamine formed in a transmethylation reaction catalyzed by the enzyme, as previously described [25]. S-adenosyl-L-(methyl-^14^C)-methionine was used as a donor of methyl group. 

The enzyme activities are expressed as pmol/min/mg protein. Protein concentration was analyzed by Lowry’s method [26].

#### 2.5.2. HPLC Detection of Monoamines and Their Metabolites in Rat Brain Tissue Samples

The concentration of dopamine (DA), serotonin (5-HT) and noradrenaline (NA) as well as their metabolites, i.e., 3,4-dihydroxyphenylacetic acid (DOPAC), homovanillic acid (HVA), 3-methoxy-4-hydroxyphenylglycol (MHPG) and 5-hydroxyindoleacetic acid (5-HIAA) was determined in striatum (STR), hypothalamus (HPT) and cerebral cortex (CTX) with the RP-HPLC-ED method. 

Cerebral samples for HPLC analysis were homogenized using an ultrasonic homogenizer (Fisher BioBlock Scientific, France) for 15 s in 150 μL homogenization solution (0.1 M HClO_4_ containing 0.4 mM Na_2_S_2_O_5_), and centrifuged at 12,000 rpm for 15 min at 4 °C. At least 100 μL of the supernatant was transferred to chromatographic tubes and kept at –80 °C until analysis. Next, 20 μL of the filtrates was injected into the HPLC system. 

The Agilent 1100 chromatographic system with Waters Spherisorb ODS-1 RP C-18 chromatographic column (4.6 × 250 mm) preceded by a Zorbax SB-C18 pre-column (4.6 × 12.5 mm) was used. Column temperature was set at 35 °C and mobile phase flow at 1 mL/min. The glassy carbon working electrode was set at +0.65 V, relative to the Ag/AgCl reference electrode. The mobile phase consisted of a phosphate buffer (3.4 pH) containing: 0.15 M NaH_2_PO_4_ x H_2_O, 0.1 M Na_2_EDTA, 0.5 mM Na_2_OSA, 0.5 mM LiCl and addition of methanol (10%). The chromatographic data were analyzed using ChemStation, Revision-B.03.02, Agilent software [27].

The concentrations of monoamines and their metabolites in each sample were calculated from the integrated chromatographic peak area and presented in nmol/gram of wet tissue. Additionally, the ratio of metabolites to their parent amines was calculated.

### 2.6. Statistical Analysis

The results were expressed as means ± standard errors of the mean (SEM). All statistical analyses were performed using GraphPad Prism 6.07 program (GraphPad Software, Inc., San Diego, CA, USA). 

The effect of pharmacological treatment was assessed with Paired *t*-test. For biochemical studies, statistical significance was determined by One-way ANOVA followed by post hoc Tukey’s or Dunnett’s multiple comparisons test.

The values *p* < 0.05 (*), *p* < 0.01 (**), and *p* < 0.001 (***) were considered significant.

## 3. Results and Discussion

### 3.1. Chemistry

Compounds **4**–**28** were synthesized as described previously [6]. First, phenoxypropyl bromides (IIa-IIl) were obtained by *O*-alkylation of a proper phenol in acetone in the presence of potassium carbonate. Such obtained compounds (IIa-IIl) were preliminarily purified and crude products were used for the reaction with proper amines as seen in Figure 1. The final compounds were purified by flash chromatography and oily products were transformed into oxalic acid salt (except 8 and 25—hydrogen chloride). Carbamates 29–32 were synthesized from the appropriate isocyanate and 3-(piperidin-1-yl)propan-1-ol as reported by Łażewska et al. [10] (Figure 2). The final compounds were purified by column chromatography and oily products were transformed into oxalic acid salt. The structures and purity of compounds were confirmed by ^1^H NMR, ^13^C NMR and LC-MS analysis (see Appendix A).

### 3.2. In Vitro Pharmacological Studies

#### 3.2.1. Histamine H_3_ Receptor Affinity of Tested Compounds

Affinity for *h*H_3_R was evaluated in a radioligand binding assay using [^3^H]*N*^α^-methylhistamine as radioligand in CHO K1 or HEK293 cells stably expressing *h*H_3_R as described previously [6,7]. Results are presented in Table 1, Table 2, Table 3 and Table 4. For comparison, DL76 (our lead structure) was tested in both assays. Results obtained for DL76 in CHO K1 cells are slightly lower (*K_i_* = 58 nM) than in HEK293 cells (*K_i_* = 38 nM), the same as our results for pitolisant (CHO K1 cells: *K_i_* = 30 nM compared with published data for HEK293 cells: *K_i_* = 12 nM [28]). At the beginning, compounds **4**–**9**, **11**, **13**–**17** and **24**–**32** were screened for the inhibition of [^3^H]*N^α^*-methylhistamine binding to the *h*H_3_R (in CHO K1 cells) at the 1 μM concentration. Then, those with at least 50% inhibition for *h*H_3_R were selected for further testing (*K_i_* evaluation). In the first series (compounds **4**–**17**; Table 1), an influence of an amine moiety for *h*H_3_R affinity was investigated. Among the acyclic amines, no correlation was observed between the length of the carbon chain (methyl to propyl; compounds **4**,**5**,**7** and **9**), or the branching (compounds **6** or **8**) for *h*H_3_R. The most potent was compound **9** with a *K_i_* of 323 nM. In the cyclic amines series (compounds **10**–**17**) the highest *h*H_3_R affinity was shown by the 2-methylpyrrolidine derivative (compound **13**) with a *K_i_* of 25 nM. A methyl substituent at the amine ring seems profitable for *h*H_3_R affinity. Compounds **10** (2-methyl) and **11** (2,6-dimethyl), derivatives of a piperidine with the methyl substituent, had very good affinity for *h*H_3_R (*K_i_* < 100 nM). Other amines (morpholine, substituted piperazines) showed no activity at all (compounds **15**, **17**) or exhibited weak potency (compounds **14**, **16**). In the second series (compounds **18**–**23**), we investigated the change of the 4-*tert*-butyl substituent at the phenyl ring for other groups: acetyl, alkyl (methyl, ethyl, isopropyl) or halogen (-Cl, -F). Compounds were tested in the binding assay in HEK293 cells [7]. All compounds showed good *h*H_3_R affinity with *K_i_* values below 100 nM. The change for an acetyl group (compound **18**) was the most profitable. Compound **18** (*K_i_*= 15 nM) was twice as active as DL76 (*K_i_*= 38 nM). Next, we performed a modification of DL76 by systematically removing methyl groups from the 4-*tert-*butyl substituent of DL76 to an isopropyl (compound **19**), an ethyl (compound **20**), and a methyl (compound **21**) (Table 2). While compounds **19** and **20** exhibited slightly lower activity (*K_i_* of 52 nM and 62 nM, respectively) than the parent DL76, compound **21** showed comparable affinity (*K_i_* of 43 nM). Interestingly, the introduction of halogen atoms (compounds **22**, **23**) resulted in decreased *h*H_3_R affinity (*K_i_* > 80 nM). Further exploration of the influence of 4-*tert-*butyl position on the phenyl ring (third series: compounds **24**–**28**; Table 3) showed that the presence of a substituent at the 4 position is very important for *h*H_3_R affinity. Compounds **24** and **25** had much lower affinity than DL76, and the 2 position was the least favorable (compound **24** with a *K_i_* > 1000 nM). In the next step, due to the good commercial accessibility of phenols, compound **24** was subsequently modified by adding a methyl substituent at the phenyl ring to obtain compounds **26**–**28**. Compound **28** with the methyl group at the 4 position had moderate affinity for *h*H_3_R (*K_i_* = 448 nM) whereas compounds with 5-methyl (**27**) and 6-methyl (**26**) were inactive (*K*_i_ > 1000 nM). Introduction of this second substituent led to an increase in potency compared to compound **24** with only 2-*tert-*butyl substituent at the phenyl ring. In the last series (compounds **29**–**32**; Table 4), the ether linker in DL76 was exchanged for a carbamate group (compound **29**). This probe led to a considerable decrease in *h*H_3_R affinity for compound **29** (*K*_i_ > 1000 nM). Next, we also changed 4-*tert*-butylphenyl moiety for aliphatic substituents with the *tert*-butyl group (compounds **30**–**32**). The resulting derivatives (**30**–**32**) exhibited no affinity for *h*H_3_R (*K*_i_ > 1000 nM). To sum up, of all investigated changes, only the replacement of the piperidine (in DL76) by a 2-methylpyrrolidine and the 4-*tert*-butyl substituent by a 4-acetyl resulted in compounds with high affinity for *h*H_3_R.

#### 3.2.2. Human MAO B Inhibitory Activity of Tested Compounds

All compounds were first screened for hMAO B inhibitory activity at the concentration of 1 μM. Then, those which showed inhibition higher than 50% were selected for further IC_50_ evaluation. The obtained results showed different abilities of tested compounds to inhibit hMAO B activity (Table 1, Table 2, Table 3 and Table 4). The most potent hMAO B inhibitors were found among compounds from the first series (Table 1). All aliphatic amine derivatives (compounds **4**–**9**) exhibited IC_50_ values in low nanomolar concentration ranges (IC_50_ < 40 nM) and a dipropyl amine derivative **9** was the most potent among them (IC_50_ of 2 nM). Moreover, among cyclic amine derivatives, very potent hMAO B inhibitors were found, with IC_50_ values ≤ 11 nM (compounds **10**–**13**, e.g., compound **13** with an IC_50_ of 4 nM). In the other series (compounds **18**–**32**; Table 2, Table 3 and Table 4), generally, all introduced changes led to inactive or only weak active compounds (with the exceptions of **19**: IC_50_ = 21 nM and **20**: IC_50_ = 70 nM). Based on the results, we selected compounds **9** and **13** for further analysis.

#### 3.2.3. Modality of Human MAO B Reversible Inhibition of Compounds **9** and **13**

For testing modality of enzyme inhibition, we used three concentrations of inhibitors that corresponded to their IC_20_, IC_50_ and IC_80_ values. Substrate (p-tyramine) was used at concentrations: 0.05, 0.1, 0.5, 1.0, 1.5 and 2.0 mM. For compounds **9** and **13**, K_M_ and V_max_ values calculated from Michaelis–Menten curves showed behavior typical for noncompetitive inhibition (V_max_ decreased curvilinearly along with the increase in inhibitor concentration, and K_M_ was not affected) (Table 5). On the Lineweaver–Burk double-reciprocal plot, lines representing solvent and different concentrations of the inhibitor intersect to the left side of the y-axis and on the x-axis, suggesting a pure noncompetitive behavior. Inhibitors show noncompetitive modality when having equal affinity for both free enzyme and enzyme-substrate complex [29] (Figure 3 and Figure 4). In the same assay conditions, safinamide showed behavior characteristic for mixed inhibition: V_max_ decreased curvilinearly and K_M_ increased curvilinearly with the increase in the inhibitor concentration, and lines on the Lineweaver–Burk plot intersect to the left of the y-axis and above the x-axis (Table 5). This behavior suggested that the inhibitor can bind to both free enzyme and enzyme-substrate complex but with higher affinity to the free enzyme [29].

For MAO B inhibition and using p-tyramine as substrate, compounds **9** and **13** showed more promising behavior than safinamide. In the human body where substrates for MAO B are present and their concentration changes, the equal affinity to free enzyme and enzyme-substrate complex could prove to be an asset.

#### 3.2.4. Reversibility of Monoamine Oxidase B Inhibition of Compounds **9** and **13**

Curves on the Figure 5A,B represent the reactivation of the MAO B activity after the addition of the excess amount of the substrate (p-tyramine 200 uM) to the enzyme that had been firstly inhibited by reference and tested compounds in concentrations corresponding to their IC_80_. Irreversible inhibition by rasagiline was clearly shown as the line that represents the amount of the product of the MAO B remained flat even after the addition of the excess amount of the substrate. In contrast, for the lines that represent safinamide, **9** and **13** showed an increase in the product amount with increase in substrate concentration which suggested reversible inhibition. Additionally, comparing the curves for two variants of the reversibility testing with and without preincubation (Figure 5C,D), safinamide and compounds **9** and **13** did not show differences between the variants which suggested very quick inhibition (i.e., noncovalent bonding), while preincubated rasagiline inhibited the enzyme more strongly than non-preincubated (as irreversible and mechanism-based inhibitor rasagiline requires time to be metabolised by MAO B to its reactive form which then forms covalent bonds with the enzyme [30]). 

#### 3.2.5. Permeability of Compounds **9** and **13**

For compounds acting on the CNS, the ability to cross the blood–brain barrier (BBB) is very important. It is good to assess this property before starting in vivo studies. Therefore, the permeability of the two most potent hMAO B inhibitors (compound **9** and **13**) was assessed using the Parallel Artificial Membrane Permeability Assay (PAMPA). This commercially available method assesses the passive transport of compounds. The assay is performed in multiwell microplates, which consist of an acceptor part and a donor part, separated by a lipid-saturated microporous filter. The results of the test are summarized in Table 6. Only for the compound **13** was it possible to calculate the permeability coefficient (*P_e_*). The results showed that the compound **9** was not able to cross the artificial membrane, as no mass peak of the compound was observed in the acceptor part. In contrast, the compound **13** had a high permeability, as the calculated *P_e_* (*P_e_* = 16.72 × 10^−6^ cm/s) was very high and comparable to caffeine (*P_e_* = 15.1 × 10^−6^ cm/s).

#### 3.2.6. Effect of Compounds **9** and **13** on the Viability of Human Astrocyte Cell Lines

In the next step, we investigated the effect of two of the most promising hybrids, compounds **9** and **13**, on the viability of human astrocyte cell lines after 24 h and 72 h of incubation. Pitolisant, the known H_3_R ligand, was used as a reference drug [17]. The examined compounds were applied in 8 concentrations (from 0.01 mg/mL to 0.25 mg/mL). Their effects on the viability of astrocytes after 24 h of incubation are presented in Figure 6. According to the obtained data, the two lowest concentrations of tested compounds (i.e., 0.01 and 0.025 mg/mL) did not affect cell viability. Regarding the successive doses of the agents used, a dose-dependent decrease in cell survival was observed, which was statistically significant. The highest decline in cell viability, over 95% in comparison to the control level, was observed for compounds **9** and **13** at the concentration of 0.15 and 0.25 mg/mL.

Interestingly, the threefold extension of the incubation time with the tested compounds in the same concentration range resulted in only a slight increase in cytotoxicity. Human astrocytes after 72 h of incubation with compound **9** and **13** as well as pitolisant were characterized by a similar survival rate to that during exposure to the test agents for 24 h (Table 7). In general, MTT conversion tests performed on human astrocyte cell lines showed slightly higher toxicity of compound **9** and **13** compared to pitolisant, which is documented by the calculated IC_50_ values (Table 7), with higher values indicating lower cytotoxicity reported for compound **13**.

### 3.3. Preliminary Verification of In Vivo Activity of the Compound **13**

The presented research focused on the search for new multifunctional compounds combining the properties of an MAO B inhibitor and the H_3_R. Taking into account the results of in vitro studies on the affinity for *h*H_3_R and hMAO B inhibitory activity (*h*H_3_R: *K_i_* = 25 nM; hMAO B: IC_50_ = 4 nM; Table 1) as well as low cytotoxicity (Table 7), and predicted very good in vivo permeability in the PAMPA assay (Table 6), the compound **13** was selected for in vivo studies. Thus, if H_3_R antagonists cross the BBB, they should affect food intake [18,31]. Experiments were conducted as described previously [20,32,33]. 

The assessment concerned the effects of compound **13** on the feeding behavior of rats after its repeated peripheral injections, and the influence on metabolism and concentration of selected key neurotransmitters. Pitolisant was used as the reference drug in in vivo studies [17]. 

#### 3.3.1. Effect on Sub-Chronic Administration of Compound **13** on Feeding Behavior 

The effect of sub-chronic administration of compound **13** and pitolisant on food and water consumption as well as urine output is presented in Figure 7. 

In the compound **13**-treated group, a statistically significant decline in food consumption was noted, compared with the results obtained before the drug’s administration (8.61 ± 0.11 vs. 10.63 ± 0.37 g/100 g bw; paired *t*-test, *p* < 0.001). Compound **13** affected the feeding pattern more than pitolisant (8.95 ± 0.14 vs. 10.57 ± 0.42 g/100 g bw; paired *t*-test, *p* < 0.01). In addition, rats injected with compound **13** had lower water consumption, expressed in mL/100 g bw/24 h (13.37 ± 0.31 vs. 15.035 ± 0.37; paired *t*-test, *p* < 0.05) and decreased urine output (8.90 ± 0.045 vs. 10.08 ± 0.35 mL/24 h; paired *t*-test, *p* < 0.01), compared to the pre-treatment period. 

Subcutaneous administration of pitolisant to rats for 6 days also resulted in reduced water consumption and urine excretion, although these changes were not statistically significant (14.16 ± 0.43 vs. 15.39 ± 0.43 mL/100 g bw/24 h and 9.08 ± 0.49 vs. 9.98 ± 0.44 mL/24 h, respectively). 

In the control group, which was administered 0.2 mL of physiological saline, no changes in the consumption of feed and water nor in urine excretion were observed in both tested time intervals. 

#### 3.3.2. Activity of MAOs and HNMT in Rat Cerebral Cortex after Sub-Chronic Administration of Compound **13**

In the concentration used, compound **13** caused more than 90% decline in MAO B activity in rat cerebral cortex (Figure 8), whereas MAO A activity was inhibited only by 12% (data not shown).

The activity of MAO B was significantly reduced after administration of compound **13** at a dose of 3 mg/kg/day for 6 days, compared to the control group (48.38 ± 12.02 vs. 584.50 ± 19.14 pmol/min/mg protein; one-way ANOVA and Tukey’s multiple comparisons test, *p* < 0.001). In the pitolisant-treated group, MAO A and B activities were close to that recorded in control animals.

Compound **13** did not influence HNMT activity. Similar activity of HNMT was noted in all studied groups, i.e., in the compound **13** group—40.09 ± 1.77, in the pitolisant group—42.51 ± 1.40, and in the control group—37.53 ±1.22 pmol/min/mg of protein.

#### 3.3.3. Effects of Sub-Chronic Administration of Compound **13** on Cerebral Concentration of Monoamines and Their Metabolites

In the compound **13** group, a statistically significant increase in DA content in CTX and STR was noted, compared with the results obtained for control animals (CTX: 1.777 ± 0.128 vs. 1.125 ± 0.145 nmol/g wet tissue, STR: 9.914 ± 1.718 vs. 5.244 ± 0.617 nmol/g wet tissue; Figure 9A,C, left panel). In contrast to these results, six-day subcutaneous administration of compound **13** caused a decrease in DOPAC levels in CTX and STR (CTX: 0.562 ± 0.093 vs. 0.857 ± 0.029 nmol/g wet tissue, STR: 1.704 ± 0.187 vs. 2.934 ± 0.296 nmol/g wet tissue; Figure 9A,C, right panel). The concentration of DA and DOPAC in these brain structures correspond with a decline in DOPAC/DA ratio. Moreover, a decrease in the HVA/DA ratio in CTX was noted (Table 8).

In the case of the hypothalamus, a slight increase in DA concentration and a decrease in DOPAC concentration were noted in the compound **13** group, although these changes were not statistically significant (Figure 9B).

Additionally, injections of compound **13** also slightly increased NA concentration in CTX (from 1.166 ± 0.084 to 1.255 ± 0.152 nmol/g wet tissue) and significantly decreased MHPG concentration (from 1.659 ± 0.048 to 0.897 ± 0.280 nmol/g wet tissue), expressed as a lower MHPG/NA ratio (0.79 ± 0.28 vs. 1.47 ± 0.13). These results are presented in Figure 10A and Table 8, respectively. In the other examined brain structures, no differences were found in the concentration of NA and MHPG (Figure 10B,C).

Contrary to catecholamines, sub-chronic administration of compound **13** caused a statistically significant reduction in the concentration of 5-HT and 5-HIAA in the cerebral cortex, relative to both the control and pitolisant-treated rats. HPLC analysis showed that the level of 5-HT in CTX in the compound **13** group was 1.194 ± 0.139 compared to 1.931 ± 0.167 nmol/g wet tissue in the the control group. Regarding the serotonin metabolite, 5-HIAA, the following values were obtained: 1.320 ± 0.153 and 2.246 ± 0.060 nmol/g wet tissue for the compound **13**-treated animals and control rats, respectively (Figure 11A).

In addition, there was a statistically significant decrease in the level of 5-HIAA in the hypothalamus, with no changes in 5-HT content (Figure 11B).

Compound **13** did not affect the 5-HT and 5-HIAA concentrations in the striatum (Figure 11C).

In the group of rats injected with pitolisant, post-mortem assays did not show any differences in the content in brain tissue of the examined biogenic amines nor in their metabolites compared to the control animals (Figure 9, Figure 10 and Figure 11, Table 8).

## 4. Discussion

According to epidemiological data, PD is the second most common neurodegenerative disorder worldwide. Inadequacies of the current pharmacotherapies to treat PD prompt efforts to identify novel drug targets. New therapeutic strategies comprise multifunctional drugs. It is assumed that drugs combining more than one activity desired in the treatment of PD will be more effective than monotherapy.

The presented research aimed to derive compounds that effectively block MAO B and show high affinity for H_3_R. Continuing our previous works in this field, analogues of the compound DL76 (1-(3-(4-tert-butylphenoxy)propyl)piperidine, dual target ligand (hH_3_R: *K*_i_ = 57 nM; hMAO B IC_50_ = 48 nM) were designed and synthesized [10,11,13]. All compounds obtained were tested for affinity to *h*H_3_R stably expressed in CHO or HEK293 cells as well as for inhibitory activity against hMAO B [6,11]. The evaluated compounds showed different activity towards both biological targets. Most of them had weak affinity for *h*H_3_R (*K_i_* > 500 nM), but very good inhibitory potency for hMAO B (IC_50_ < 50 nM). The most promising dual-acting ligand appeared to be 1-(3-(4-(*tert*-butyl)phenoxy)propyl)-2-methylpyrrolidine (compound **13**) (*h*H_3_R: *K_i_*= 25 nM; hMAO B IC_50_ = 4 nM) whereas compound **9** (3-(4-(*tert*-butyl)phenoxy)-*N,N*-dipropylpropan-1-amine) was the most potent hMAO B inhibitor (IC_50_ = 2 nM) with moderate affinity for *h*H_3_R (*K_i_*= 325 nM). Both compounds were selected for further in vitro studies. Kinetic evaluation of hMAO B inhibition showed noncompetitive and reversible behavior of both compounds. To our surprise, in the PAMPA assay, differences in penetration of the compounds through the artificial membrane were observed. Compound **9** did not penetrate while compound **13** had a high penetration capacity. The permeability of a molecule across the cell membrane is an important factor determining the oral absorption and bioavailability of a drug. The lack of permeation of compound **9** is not easy to explain. 

First, the experiment was repeated, as the result obtained surprised us. However, both tests gave the same result. Since the permeation of PAMPA is influenced by the chemical structure of the molecule, physicochemical parameter calculations (SwissAdme program: http://www.swissadme.ch; accessed on 22 February 2022) were performed to check this. The calculations were carried out for compound **9** and compound **13**. The results obtained, however, showed no significant differences in the values of these parameters (slightly higher logP of compound **9**—4.86 vs. 4.11 for compound **13**). One significant difference was the number of rotational bonds in the molecule (10 bonds for compound **9** and 6 bonds for compound **13**). It is known that in addition to molecular weight, the flexibility of the molecule (measured by the number of rotational bonds, polar surface area or total number of hydrogen bonds, i.e., the sum of donors and acceptors) is an important predictor of good oral bioavailability. In this case, the differences in molecular weight between compound **9** (MW=291 g/mol) and compound **13** (MW =275 g/mol) are small, and the TPSA (12.47 Å2) and the number of hydrogen bonds (2) are the same. Thus, it is likely that the molecular flexibility of compound **9** determines its permeability, but this requires further research to confirm. 

Further in vitro studies of both tested compounds (**9** and **13**) showed a dose-dependent decrease in the viability of the human astrocytes from the cerebral cortex, which was similar after 24 h and 72 h [Figure 6, Table 7]. Thus, the results of all in vitro studies allowed us to select a promising compound **13** for in vivo evaluation.

Experimental and preclinical studies performed on different animal models have convincingly shown that H_3_Rs play an important role in energy balance and body weight gain and their antagonist/inverse agonists act as anorexic drugs [32,33,34]. Based on these reports, it was assumed that if a tested compound administered peripherally crosses the BBB, and has an antagonistic affinity for H_3_Rs, it should inhibit food intake. Pitolisant, a H_3_R antagonist/inverse agonist [17], was used as a reference compound. Assessment of feeding behavior in rats was performed in metabolic cages that allow precise control of daily feed and water consumption as well as urine output. As expected, in vivo studies showed that compound **13** (administered subcutaneously) crosses the BBB and inhibits feed consumption in rats to an extent similar to pitolisant (Figure 7). This observation confirms that compound **13** exhibits typical effects on feed consumption for an H_3_R antagonist/inverse agonist.

In PD therapy, it is especially important to raise the cerebral DA level. MAO B inhibitors may increase DA availability in PD brain. Experimental data also suggest that MAO B inhibitors act as neuroprotective agents by decreasing the production of potentially dangerous by-products of DA metabolism in the brain [35]. In addition to symptomatic effects caused by MAO B inhibitors, it is also worth noting that (1) post-mortem analysis showed an age-related increase in MAO B activity in the human brain [36] and (2) the enzyme is also located in the glial cells, so its enhanced activity may be a result of age-associated glial cell proliferation [35]. Thereafter, the cerebral activity of MAOs as well as concentrations of catecholamines, serotonin and their key metabolites were studied post-mortem in rats treated with compound **13**. The examined compound turned out to be a very effective MAO B inhibitor. Subcutaneous administration of it to rats for 6 days at a dose of 3 mg/kg body weight reduced MAO B activity by more than 90% (Figure 8). This is further evidence that compound **13** crosses the BBB. Post-mortem biochemical analyses in animals treated with compound **13** also showed a higher concentration of DA in the striatum and cerebral cortex (Figure 9A,C). This alteration was associated with a decrease in the concentration of DOPAC, the direct product of DA deamination by MAO B. Thus, the observed result was caused primarily by the blocking of MAO B activity by the tested compound. The correctness of this thesis was proved by a decline in DA turnover expressed as the decreased DOPAC/DA ratio (Table 8). In addition, sub-chronic injections of compound **13** caused a slight increase in NA concentration in the cerebral cortex and a statistically significant decrease in the concentration of MHPG, the final product of amine inactivation through combined deamination and methylation processes (Figure 10A). A decrease in the NA turnover was also expressed by a lower MHPG/NA ratio. It seems that the reduced MHPG concentration (and thus the lower MHPG/NA ratio value) was also a consequence of the diminishing MAOs activity. On the other hand, the tendency to increase the concentration in NA may also be partly due to a rise in DA level (its immediate precursor).

As already mentioned, the H_3_R also act as heteroreceptors which modulate the release of other neurotransmitters [17,18]. At this stage of the studies, it cannot be ruled out that the increase in catecholamine levels, especially DA, may also be the result of antagonistic activity of compound **13** towards H_3_R. 

Surprisingly, animals treated with compound **13** had a lower concentration of serotonin (5-HT) in the cerebral cortex, relative to both the control and pitolisant-treated groups. The decrease in the level of 5-HT was also accompanied by a reduced concentration in 5-HIAA, the final amine’s metabolite (Figure 11A). Turnover of 5-HT is comparable to that recorded in both other groups of rats (Table 8). The interpretation of this phenomenon requires further investigations.

Serotonin is mainly metabolized by MAO A [37,38], while administration of compound **13** lowered MAO A activity by only 12%. The concentration in 5-HT in the brain is the result of synthesis, release, reuptake, and regulation by auto- and heteroreceptors, as well as the influence of other factors that are difficult to define [39,40]. 

Particularly noteworthy is the 5-HT_1A_ receptor due to its key role in the autoregulation of the brain 5-HT system functional activity. Stimulation of serotonin 5-HT_1A_ receptors (5-HT_1A_Rs) leads to reduction in the neuronal firing rate [41,42]. According to localization, the 5-HT_1A_Rs are powerful regulators of both pre- and postsynaptic 5-HT neurotransmission. HT_1A_Rs are found on 5-HT cell bodies and dendrites, mainly in the midbrain raphe nucleus region (presynaptically located autoreceptors) and on terminal targets of 5-HT release (postsynaptic 5-HT_1A_ receptors). 5-HT_1A_ autoreceptors inhibit neuronal spike activity in dorsal raphe nucleus and 5-HT release into the synaptic cleft [39,41]. Postsynaptic 5-HT_1A_Rs receptors mediate the action of 5-HT on neurons and also could regulate 5-HT system functional activity via complex feedback neural networks [43,44]. 

In subsequent studies, we will try to verify the effect of the compound **13** on the brain’s serotoninergic system, including its binding affinity for 5-HT_1A_ receptors.

## 5. Conclusions

Among all designed compounds, we were able to obtain compound **13**, a dual ligand, with high affinity for *h*H_3_R and strong inhibition of hMAO B. The in vivo studies performed confirmed its ability to cross the BBB and showed typical effects on feed consumption for an H_3_R antagonist. Moreover, compound **13** strongly inhibited brain activity of MAO B with little effect on inhibition of MAO A. Furthermore, these studies showed a positive effect on increasing cerebral DA levels in the rat’s brain. In conclusion, the results presented here predispose this compound to further experimental studies to assess its full therapeutic potential in PD.

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
