# Peer review of "Dual Targeting Ligands—Histamine H3 Receptor Ligands with Monoamine Oxidase B Inhibitory Activity—In Vitro and In Vivo Evaluation"

_pharmaceutics, 2022, doi:10.3390/pharmaceutics14102187_

Round 1

Reviewer 1 Report

Combination of classical targets hMAO B and new targets histamine H3R is a novel strategy for treatment of Parkinson’s disease. The authors have previously synthesized dual target ligands blocking hMAO B and hH3R which showed higher inhibitory activity for hMAO B. In this paper, the most potent hMAO B inhibitors (9 and 13) were screened and assessed for hMAO B and histamine H3R inhibition and the ability to cross blood-brain barrier. The compound 13 was selected for in vivo test which could cross the BBB and showed a positive effect on increasing cerebral DA levels in the rats’ brain.

This study provided potential new therapeutic agents for treating Parkinson’s disease by dual targeting hMAO B and histamine H3R. The research procedure was well designed and the experiment methods were well described. The results were clearly presented and properly discussed. The writing is fluent and meets the standard for acceptable for the journal.

Several minor comments are suggested as follow:

First, the abstract could include a conclusion. Second, in figure 9, the A, B, C may be aligned. Third, in line 381, the “IC50” should not be the Italic type.

Reviewer 2 Report

In general it is a great manuscript, the results found are too good, the scientific rigour with which the different tests were carried out is evident throughout the research process. However, I have some comments:

* In Figure 1 the compound DL76 does not appear or is not indicated.

* Compounds 9 and 13 are mentioned in the introduction but their structure is not given. In this same way, I recommend to make a table with the structures that were obtained, I consider that it would facilitate the reading of the manuscript.

* In the methodology it is stated "radioligand binding assay as described previously" but there was no description as such.

* Scheme 1 is not mentioned in the manuscript.

* In the viability assays, it is stated that: "In general, MTT conversion test performed on human astrocyte cell line showed 711 lower toxicity of compound 9 and 13 compared to pitolisant..." but the results show that both compound 9 and 13 showed higher toxicity. It is recommended to expand on this idea.

* In the discussion, the in vivo results are very well analyzed, but the in vitro results refer as such to the result but not to the possible phenomenon, for example: why compound 9 did not penetrate in the membrane penetration assays? It is recommended to analyze the in vitro results in more detail.
